# Effect of Ethanol on Exosome Biogenesis: Possible Mechanisms and Therapeutic Implications

**DOI:** 10.3390/biom13020222

**Published:** 2023-01-24

**Authors:** Vaishnavi Sundar, Viswanathan Saraswathi

**Affiliations:** Division of Diabetes, Endocrinology and Metabolism, Department of Internal Medicine, University of Nebraska Medical Center and VA Nebraska-Western Iowa Health Care System, Omaha, NE 68105, USA

**Keywords:** exosomes, hepatocytes, alcohol-associated liver disease, ethanol, cirrhosis

## Abstract

Most eukaryotic cells, including hepatocytes, secrete exosomes into the extracellular space, which are vesicles facilitating horizontal cell-to-cell communication of molecular signals and physiological cues. The molecular cues for cellular functions are carried by exosomes via specific mRNAs, microRNAs, and proteins. Exosomes released by liver cells are a vital part of biomolecular communication in liver diseases. Importantly, exosomes play a critical role in mediating alcohol-associated liver disease (ALD) and are potential biomarkers for ALD. Moreover, alcohol exposure itself promotes exosome biogenesis and release from the livers of humans and rodent models. However, the mechanisms by which alcohol promotes exosome biogenesis in hepatocytes are still unclear. Of note, alcohol exposure leads to liver injury by modulating various cellular processes, including autophagy, ER stress, oxidative stress, and epigenetics. Evidence suggests that there is a link between each of these processes with exosome biogenesis. The aim of this review article is to discuss the interplay between ethanol exposure and these altered cellular processes in promoting hepatocyte exosome biogenesis and release. Based on the available literature, we summarize and discuss the potential mechanisms by which ethanol induces exosome release from hepatocytes, which in turn leads to the progression of ALD.

## 1. Introduction

In the year 2017, more than 2 million individuals were affected by alcohol-associated liver diseases (ALD) in the US, and it has been rapidly becoming a global health burden that requires immediate medical care [1]. ALD is also one of the predominant causes of liver-associated mortality, resulting in the higher demand for liver transplant in high-income economies [2]. The temporal evolution of ALD-associated pathology begins with early steatosis, followed by liver inflammation, necrosis, and subsequent fibrosis, eventually leading to liver cirrhosis and hepatocellular carcinoma [3]. While the liver is the primary target for ethanol-induced injury due to its role in ethanol metabolism, many other organs, including the gut, pancreas, brain, and heart are also vulnerable to ethanol-induced injury [4]. Ethanol exerts direct effects in these organs, leading to their dysfunction. In addition, the altered organ function due to chronic ethanol exposure impairs the functions of other distal organs by organ–organ crosstalk [5]. For example, the adipose–liver, and gut–liver axes have been known to play critical roles in mediating the progression of ALD [6,7].

It is firmly established that within an individual organ the effect of ethanol in one cell type is transmitted to another cell type, thereby propagating ethanol-induced organ injury. With regards to ALD, ethanol’s effects in hepatocytes can alter the functions of Kupffer cells, which in turn escalates the inflammatory response of the latter [8,9].

Likewise, ethanol-induced hepatocyte injury is known to aberrantly regulate the functions of hepatic stellate cells (HSCs), thereby leading to liver injury [10,11]. Thus, several types of ethanol-induced changes in hepatocytes affect the functions of other cell types in liver. Conversely, the nonparenchymal cells, including Kupffer cells, HSCs, and endothelial cells, are also known to interact with each other and with hepatocytes, leading to the development of alcohol and non-alcohol-related liver injury [12,13,14,15,16,17]. 

Ethanol’s effects on the gut microbiome can also influence the functions of liver cells, leading to liver injury [18,19,20,21]. For example, ethanol-induced alterations in intestinal microflora activity can cause intestinal barrier dysfunction [22,23]. The gut microflora secretes endotoxins such as lipopolysaccharide (LPS) and β-glucan that can easily translocate into the liver by crossing the damaged intestinal barrier, hastening the progression of ALD [24]. In particular, LPS is known to activate hepatic macrophages, leading to a substantial increase in reactive oxygen species (ROS), proinflammatory cytokines, and chemokines, finally resulting in liver injury [15,25]. In addition to macrophages, other nonparenchymal cells as well as parenchymal cells are also activated by LPS, thereby contributing to the severity of liver disease [26,27,28,29]. Additionally, it was found that alcohol accelerates the progression of liver damage, leading to fibrosis and cirrhosis in patients coinfected with human immunodeficiency virus (HIV) or hepatitis B virus [30].

Emerging evidence suggests that ethanol-induced extracellular vesicles (EVs) play a critical role in promoting the pathological cell–cell interactions that lead to ALD [17,31,32]. With the advent of new technologies to isolate and detect exosomes, researchers have been focusing on elucidating the exosome function in the pathological progression of ALD. Exosomes are extracellular nanovesicles of nearly 40 nm to 160 nm that carry biological information via lipids, proteins, and coding and noncoding RNAs [33]. Historically, an exosome was thought to be a cellular waste removal system, but later it was widely accepted that it is a membrane-derived structure originating from multivesicular bodies of the endosomal pathway [34]. A vesicular body is formed by the invagination of the plasma membrane and the endocytosis of proteins, lipids, mRNAs, and microRNAs [33]. After maturation, some multivesicular bodies fuse with the plasma membrane and subsequently release intraluminal vesicles (ILVs) called “exosomes” into the extracellular space [33]. Further, exosomes act as cellular “shuttles”, carrying biomolecules, communicating between different cell types. Exosomes are utilized as a strategy for cellular communication in regulating organ homeostasis under normal conditions [35,36]. Under normal physiological conditions, hepatocytes release exosomes to restore organ integrity and homeostasis [37,38].

However, it should be noted that exosomes not only act as mediators in liver physiology but are also used as communication agents in the pathogenesis of liver disease when triggered by an external stimulus such as alcohol [39,40]. Importantly, alcohol consumption increases exosome release in various liver cell types, and these exosomes/EVs orchestrate cell–cell communication by horizontally shuttling genetic information from a donor cell to a target cell, thereby leading to ALD [40]. The possible interactions among different cell types in the liver via exosomes that lead to the pathogenesis of ALD are shown in Figure 1.

Although a great deal of effort has been devoted to understanding the mechanisms of exosome cargo loading, very little is known about the cellular cues that regulate the synthesis and secretion of exosomes. Below, we discuss the possible mechanisms by which ethanol promotes exosome biogenesis and release from hepatocytes, which in turn can lead to the pathogenesis of ALD.

## 2. Liver Exosomes in Tissue Homeostasis and Tissue Injury

The physiological functioning of the liver is achieved by a complex cross-communication between parenchymal (hepatocytes) and nonparenchymal cells (stellate cells, liver sinusoidal endothelial cells, Kupffer cells, and cholangiocytes). The hepatocytes perform most liver functions, and the nonparenchymal cells release various metabolites and molecules to support and aid hepatocytes as well as their nonparenchymal neighbors [38]. To maintain the functional order and homeostasis in the hepatic environment, these cells utilize one or more methods to communicate with each other. Exosomes provide a cellular communication strategy by conveying molecular cues to target cells at both short and long distances [38]. A prevailing misconception is that the exosomal ability to carry a variety of biological cargoes from hepatocytes to other cells or *vice versa* is pathological. However, it is important to understand that it is also a part of the homeostatic response to injury [41]. For instance, exosomes derived from primary hepatocytes in a culture containing sphingosine kinase 2 promoted liver regeneration in two-thirds partial liver resections [41]. Similarly, nonparenchymal cells, including HSC, liver sinusoidal endothelial cells, and cholangiocytes, also secrete exosomes to regulate liver remodeling upon liver injury [42]. For example, the HSC-derived exosomes create a favorable microenvironment for the release of profibrogenic factors and collagen deposition to maintain organ integrity in response to liver injury [42]. On the other hand, exosomes containing miR-214 produced by quiescent HSCs inhibit HSC activation, thereby limiting the injury response [43]. 

During a prolonged insult, exosomes released by liver cells augment the extent of liver injury. For example, Devhare et al. showed that exosomes from HCV-infected hepatocytes contain miR-19a, which targets SOCS3 to activate STAT3-mediated TGF-β signaling in HSCs and promotes HSC proliferation [44]. Additionally, damaged cholangiocytes secrete exosomes containing the long noncoding RNA H19, which conversely causes self-injury to hepatocytes due to faulty intercellular communication [45,46]. Together, healthy or injured hepatocytes release exosomes, and these exosomes have important roles in tissue homeostasis or tissue injury. However, recurrent injuries to hepatocytes upon prolonged exposure to ethanol can alter exosomal release and exosomal cargo, which in turn can aggravate liver damage.

## 3. Role of Ethanol in Exosome Release

Exosomes are persistently released into the surroundings by a variety of cells, irrespective of whether they are in a natural environment or artificially cultured. Several studies have shown that ethanol increases circulating exosomes in rodents as well as human subjects with alcohol use disorder (AUD). For example, binge and/or chronic alcohol use increases the number of circulating exosomes in healthy human subjects and in mice [40,47]. Momen-Heravi et al. showed that ethanol increases exosome release into the serum of mice as well as human subjects with AUD [47]. Further, an increased number of circulating EVs with high levels of miR-27 were found in the plasma of patients with alcohol-associated hepatitis [48]. Cho et al. showed that the exosome numbers were increased in the serum of patients with AUD, as well as in the binge ethanol-fed mice and rats, in a CYP2E1-dependent manner [49]. The increase in circulating exosomes upon ethanol exposure could be due to the ethanol’s effects on various types of cells.

Several lines of evidence suggests that ethanol increases exosome release in different cell types including hepatocytes and microglia [50,51,52]. Moreover, ethanol exposure increases exosome release from cardiomyocytes, and ROS were implicated in mediating this response [53]. Next, it was noted that the total number of EVs secreted from alcohol-treated monocytes was significantly increased compared with untreated monocytes [48]. It should be noted that hepatocytes are a significant source of exosomes upon ethanol exposure. Ethanol has been shown to increase exosome release from HepG2 hepatoma cells expressing CYP2E1 (HepG2^Cyp2E1^ cells) [54]. In the same study, the authors demonstrated that alcohol-induced EV release stimulated macrophage activation and the release of inflammatory cytokines. [54]. Momen-Heravi et al. showed that ethanol increased the number of exosomes in a dose-dependent manner in Huh 7.5 hepatoma cells and primary human hepatocytes, and these exosomes contained miRNA-122 [40]. These studies have demonstrated that ethanol triggers exosome release in hepatocytes, which in turn can mediate the development of ALD. The mechanisms by which ethanol-induced hepatocyte exosomes mediate ALD pathogenesis is beyond the scope of this review, and the reader is referred to an excellent review published on the role of ethanol-induced EVs in mediating cell–cell communication, thereby resulting in ALD [17].

## 4. Mechanisms by Which Ethanol Promotes Hepatocyte Exosome Biogenesis and/or Release

Several types of Ras-associated binding (Rab) proteins are involved in the biogenesis of exosomes. In particular, Rab11, Rab 27, and Rab 35 are directly involved in exosome biogenesis and secretion [55]. In addition, vesicle-associated membrane proteins (VAMP) and syntaxins are involved in exosome release [56,57]. Interestingly, Bala et al. showed that the expression of proteins of the Rab family such as Rab1a, Rab5c, Rab6, Rab10, Rab11, Rab27a, and Rab35 were increased at the mRNA level in primary human hepatocytes after alcohol treatment [50]. Moreover, Rab5, Rab6, and Rab11 showed significant induction in the livers of patients with ALD. Additionally, VAMP3, VAMP5, VAMP-associated protein B (VAPb), and syntaxin16 mRNA transcripts were increased in alcohol-treated cells and in the livers of alcohol liver disease patients [50]. An alcohol-induced increase in these genes was associated with increases in exosome secretion in alcohol-treated hepatocytes [50]. However, the underlying cellular mechanisms by which ethanol alters these players, leading to an increase in exosome biogenesis/release, are still unclear. Below, we discuss some of the mechanisms involved when ethanol-induced hepatocyte injuries trigger exosome synthesis/secretion.

## 5. Possible Roles of Cellular Processes Involved in ALD in Mediating Hepatocyte Exosome Biogenesis/Secretion

There are four major cellular processes that are altered in hepatocytes upon ethanol exposure, leading to ALD: (1) autophagy, (2) ER stress, (3) oxidative stress, and (4) epigenetic changes. In fact, ethanol is known to inhibit autophagy [58] and induce endoplasmic reticulum (ER) stress [59], oxidative stress [60], and the acetylation of DNA [61]. Interestingly, these pathways are also linked to exosome release in other pathological liver conditions [62,63,64]. The existing evidence suggests that changes in these cellular processes by ethanol are involved in facilitating exosome biogenesis/secretion upon ethanol exposure. Below, we summarize the roles of these individual processes in mediating ethanol’s effect in promoting exosome release.

### 5.1. Ethanol, Autophagy, and Exosomes

Autophagy is a common metabolic process in most eukaryotic cells and functions to promote cell survival. Under various stress signals, such as starvation, hypoxia, or endoplasmic reticulum stress, autophagy can degrade soluble proteins and other organelles into amino acids in the cytoplasm for energy production and biosynthesis. In addition, autophagy clears denatured or misfolded proteins and aged or damaged organelles to maintain intracellular homeostasis [65]. Under severe or chronic stress, excessive or insufficient autophagy can lead to the accumulation of large amounts of self-degradation or toxic substances, ultimately leading to cell death, which is closely associated with the pathogenesis of liver diseases [66].

Evidence suggests that there is an intricate link between exosome biogenesis and macroautophagy [67]. The selective removal and secretion of harmful proteins in exosomes or by the autophagy–lysosomal pathway are coordinated processes that participate in protein homeostasis and contribute to the maintenance of cellular fitness [67,68]. As mentioned, exosomes are formed as intraluminal vesicles (ILV) within late endosomes as a result of membrane invagination. The late endosomes, upon further maturation into multivesicular body (MVB) fuses with lysosomes, and the contents of the ILVs are degraded via autophagy [68,69]. Alternatively, when autophagy is disrupted, the MVBs fuse with the plasma membrane and release their contents into the extracellular environment as exosomes. Remarkably, autophagy modulators regulate MVB formation and exosome release [70]. Autophagy induction by starvation, rapamycin treatment, or LC3 overexpression inhibits exosome release, suggesting that, under conditions that stimulate autophagy, MVBs are directed to the autophagic pathway with the consequent inhibition of exosome release [70]. Thus, the balance between autophagy induction and exosome release are tightly regulated. However, little is known regarding the mechanistic link between exosome secretion and autophagy upon ethanol exposure.

Ethanol is known to inhibit autophagy. The importance of autophagy in ALD has been experimentally evaluated by Babuta et al. using a chronic alcohol liver disease mouse model [71]. They noted that alcohol reduced autophagy flux in vivo in chloroquine-treated mice as well as in vitro in hepatocytes and macrophages treated with bafilomycin A. Their results revealed that alcohol disrupted autophagy function at the lysosomal level through decreased lysosomal-associated membrane protein 1 (LAMP1) and lysosomal-associated membrane protein 2 (LAMP2) in livers with ALD [71]. In addition, Menk et al. also reported that chronic alcohol consumption impaired hepatocellular autophagy [72].

Exosome biogenesis and autophagy pathways are intricately linked [73], and a reciprocal relationship exists between autophagy and exosome biogenesis. For example, autophagy inhibition restores exosome release in Jurkat T cells [74]. The downregulation of LAMP1 or LAMP2, proteins involved in autophagy, increased exosome release in hepatocytes and macrophages [71]. Therefore, it is reasonable to speculate that an increase in exosome production in ALD may be linked to a disrupted autophagic mechanism. Alternatively, alcohol exposure increases exosomal miR-155, which inhibits LAMP1 and LAMP2 in the autophagy pathway [71]. This study demonstrates that miR-155-deficient mice were protected from the alcohol-induced disruption of autophagy and had attenuated exosome synthesis [71]. These studies suggest that the alcohol-induced increase in exosome production may be linked to the disruption of autophagy and impaired autophagosome and lysosome function in ALD.

### 5.2. Ethanol, ER Stress, and Exosomes

Many research studies have shown that ER stress plays a key role in the progression of nonalcoholic fatty liver disease as well as ALD [75,76]. Emerging evidence suggests that ER stress promotes exosome release in hepatocytes [77]. It has been demonstrated that the ER stress induced by palmitate in hepatocytes enhances the release of EVs [63]. Moreover, tunicamycin, an ER stress inducer, promotes the release of EVs in choriocarcinoma cells which carry death-associated molecular patterns (DAMPs), indicating the crucial crosstalk between ER stress and exosome release in regulating cell death mechanisms [78]. In addition, the treatment of HepG2 hepatoma cells with tunicamycin increased exosome secretion, and these exosomes increased PD-L1 expression in macrophages, which in turn inhibited T-cell functions [79].

However, only a few studies explored the role of ER stress in altering exosome release upon ethanol exposure. Cho et al. reported that thapsigargin, an ER stress inducer, in-creased exosome release in primary mouse hepatocytes [49]. In the same study, ethanol also increased exosome release, and this effect was blunted by 4-phenylbutryric acid, an ER stress inhibitor [49]. Further, they showed that EVs from ethanol-exposed mice, rats, and human subjects reduced the viability of primary hepatocytes. In another study, an ethanol-induced increase in exosomal miR122 was attenuated upon ER stress inhibition by 4-phenylbutyric acid [80]. These studies suggest that ER stress plays a role in promoting exosome secretion and/or cargo loading upon ethanol exposure. However, the mechanisms by which ER stress leads to exosome secretion remain unclear.

It should be noted that ER stress is critical for maintaining cell survival by activating the unfolded protein response (UPR). It is now clear that ER stress is also a potent trigger for autophagy [65]. As discussed previously, the inhibition of autophagy promotes exosome release. Thus, it is apparent that under physiological conditions the ER maintains homeostasis by utilizing the balanced action of autophagy and the exosomal release pathway or the unfolded protein response pathway. Any imbalance in these processes leads to an increase in exosome release and the pathogenesis of various diseases [81]. However, there is a huge knowledge gap in understanding the precise mechanisms leading to an imbalance in these processes for regulating exosome secretion in ALD.

### 5.3. Ethanol, Oxidative Stress, and Exosomes

Generally, mitochondria are the factories for ROS synthesis during both physiological and pathological conditions. Besides the fact that mitochondria have an intrinsic ROS scavenging ability conferred by antioxidant enzymes, including superoxide dismutase 2 and glutathione peroxidase [82], it is worth noting that this is not enough to compensate for the cellular need for ROS clearance to protect themselves from oxidative-stress-induced cell damage. Therefore, the cells deploy exosomes as an effective tool to assist with overcoming their antioxidant deficiency and protect themselves from oxidative-stress-induced cytotoxicity. For example, EVs can serve as an alternative mechanism to remove oxidized proteins after oxidative stress to maintain cellular homeostasis [83]. Oxidized lipids are also loaded into exosomes that are released from cells undergoing oxidative stress [84].

Further exploring the link between exosomes and oxidative stress in the literature reveals that exosomes from healthy cells protect target cells from oxidative injury through the transfer of antioxidants [85]. Exosomes exert their cytoprotective and anti-inflammatory properties by regulating the redox environment and oxidative stress in the liver. The antioxidant effects of exosomes in in vivo and in vitro models demonstrate that the antioxidative exosomes derived from Nrf2 over-expressing adipose mesenchymal stem cells reduce excessive ROS, inflammation, and lung injury via antioxidative stress and immunomodulation [86].

It is becoming clear that oxidative stress itself influences the release and the molecular cargo of exosomes that, in turn, injures the target cells upon ethanol exposure. CYP2E1, an enzyme involved in ethanol metabolism, is an important source of ROS. It is interesting to note that alcohol use results in increased exosome release via CYP2E1 [49]. In addition to promoting exosome release, CYP2E1 is also incorporated into the exosomal cargo in an oxidative stress-dependent manner. Moreover, the inhibition of oxidative stress using N-acetyl cysteine inhibits CYP2E1-mediated exosome release [49], indicating the role of oxidative stress in promoting exosome release. In another study, ethanol plus HIV infection triggered intense EV generation, and this was associated with an increase in oxidative stress [87]. These studies suggest that ethanol-induced oxidative stress contributes, at least in part, to exosome release.

### 5.4. Ethanol, Epigenetics, and Exosomes

Epigenetic regulation is a cellular process that brings modifications to the genome and gene expression without altering the nucleotide sequence itself. Ethanol-induced epigenetic changes including histone post-translational modifications and DNA methylation play an important role in the development of ALD [88]. Alcohol-induced histone modifications alter the expression of miRNAs in the liver, which eventually result in the pathological progression of ALD [88]. However, it is not yet known how these epigenetic changes caused by ethanol exposure modulate exosome release during ALD progression.

Epigenetic changes, particularly acetylation, play a role in exosome release. For example, lactate inhibits Sirt1 activity, thereby increasing the acetylation of high mobility group box-1 (HMGB1) in macrophages. The acetylated HMGB1 is then secreted via exosomes and increases endothelial permeability [89]. In another study, the adipose-specific knockdown of Sirt1 resulted in obesity and insulin resistance by promoting exosome release [90]. Latifkar et al. examined the consequences of depleting breast cancer cells of SIRT1. They found that reducing SIRT1 levels decreased the expression of one subunit of the vacuolar-type H+ ATPase, which is responsible for proper lysosomal acidification and protein degradation. This impairment in lysosomal function led to an increase in exosome secretion [91].

Furthermore, Lee et al. showed that, in addition to SIRT1, knockdown of SIRT2 also increased exosome release in HEK293 cells [92]. Taken together, these various reports show that the epigenetic changes caused by SIRTs inhibit exosome release and that the inhibition/loss of Sirt1 and/or Sirt2 activities result in increased exosome secretion. Chronic ethanol consumption causes steatosis and inflammation in rodents and humans, and these effects are mediated, in whole or in part, by the inhibition of SIRT 1 [61]. However, the interplay between ethanol and the SIRTs in mediating exosome release is still unknown.

## 6. Conclusions and Future Perspectives

The field of exosome biology in relation to cancer progression has been growing exponentially. It is also becoming clear that exosomes play a key role in the development of NAFLD [93]. Evidence supports a role for ethanol in increasing the formation of exosomes in cultured cells and rodent models as well as patients with alcoholic hepatitis [47]. However, little is known regarding the mechanisms by which ethanol promotes exosome biogenesis and secretion.

As depicted in Figure 2, the ethanol-mediated inhibition of autophagy and the induction of ER stress, oxidative stress, and epigenetic changes can play a role in increasing exosome formation. However, the role of other signaling pathways altered by ethanol in altering the formation of exosomes and their biological activity remains unknown. For example, ethanol inhibits AMPK signaling, and the inhibition of AMPK signaling in adipose tissue induces exosome shedding and NAFLD in mice [93]. As mentioned, ethanol is known to inhibit SIRT1, which in turn promotes the progression of ALD. Recently, SIRT1 and SIRT2 have been shown to inhibit cargo loading and the release of extracellular vesicles [92]. It would be interesting to know if an ethanol-mediated decrease in AMPK and SIRT activities in the liver plays a role in promoting exosome secretion upon ethanol exposure.

Another prevailing hypothesis in this field is that inhibiting exosome biogenesis is a potential therapeutic strategy against cancer. Accordingly, asteltoxin, a new EV secretion inhibitor, was identified and was shown to inhibit EV release via increasing AMPK signaling in cancer cells [94]. Sulfisoxazole inhibits the secretion of small extracellular vesicles by targeting endothelin receptor A in breast cancer cells [95]. Manumycin-A (MA), a natural microbial metabolite, was analyzed in exosome biogenesis and secretion in castration-resistant prostate cancer (CRPC) C4-2B cells. MA attenuated exosome biogenesis, and the inhibitory effect of MA on exosome biogenesis and secretion was primarily mediated via the targeted inhibition of Ras/Raf/ERK1/2 signaling [96]. The role of these exosome inhibitors in altering ALD remains unknown and warrants investigation. However, care should be taken in such studies, as the inhibition of exosome biogenesis has the potential to cause some adverse effects. For example, adiponectin increased exosome biogenesis and protected cells from ceramide accumulation in the cells, thereby leading to cardioprotective effects [97]. This study indicates that there are protective effects of exosomes in some conditions.

Overall, exosomes play a vital role in maintaining cellular homeostasis. It is also be-coming clear that dysregulated exosome release and their cargo contribute to the pathogenesis of a variety of diseases, including ALD. Further studies on the mechanisms by which ethanol promotes exosome secretion and the effects of inhibitors and promoters of exosome secretion will provide a mechanistic rationale for targeting these novel players to ameliorate ALD.

## Figures and Tables

**Figure 1 biomolecules-13-00222-f001:**
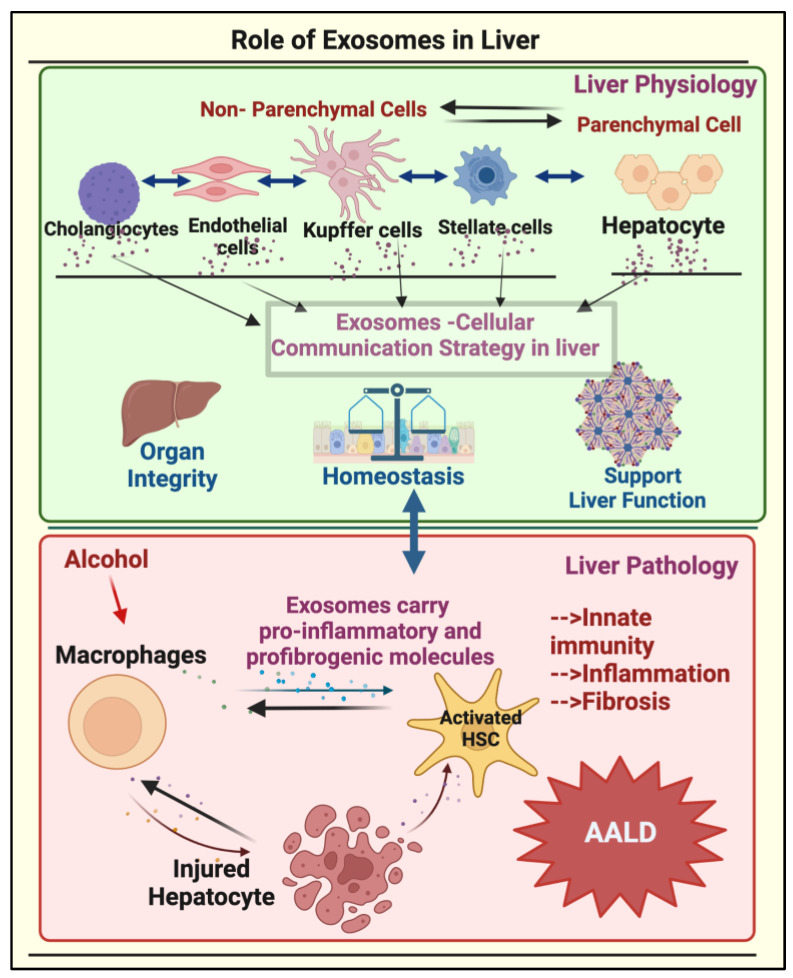
**Role of exosomes in the liver.** The liver uses exosomes as a cross-communication strategy between parenchymal (hepatocytes) and nonparenchymal cells (cholangiocytes, sinusoidal endothelial cells, Kupffer cells, and hepatic stellate cells (HSC)) for normal functioning and homeostasis. Exosomes derived from hepatocytes, macrophages, or HSCs upon ethanol exposure alter the functions of each other, leading to the progression of ALD.

**Figure 2 biomolecules-13-00222-f002:**
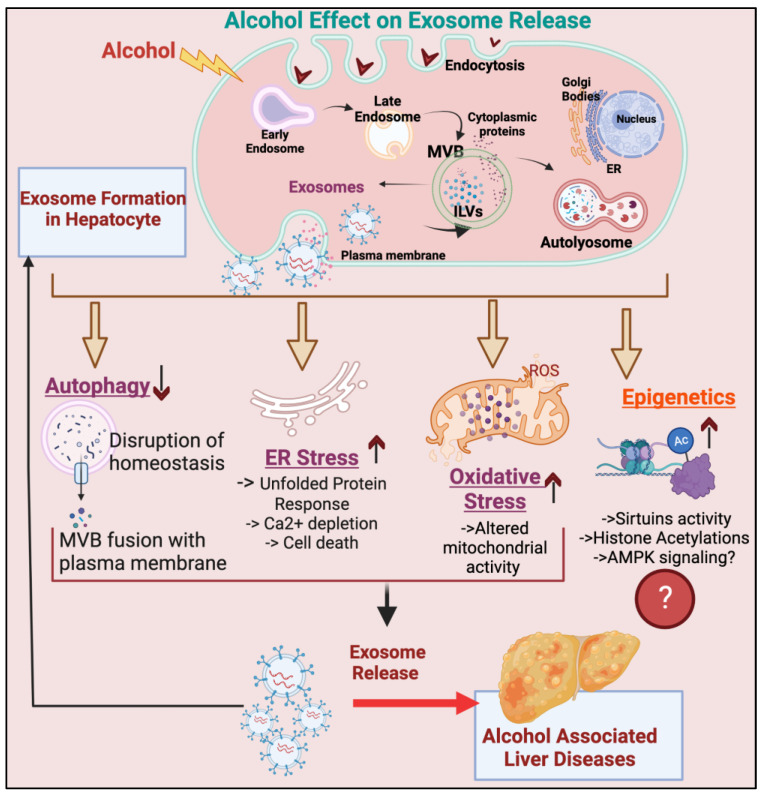
**Ethanol promotes exosome release from hepatocytes.** Exosomes are nanovesicles that are borne from the intraluminal vesicles of multivesicular bodies (MVBs) and released into fluids of the circulatory system to facilitate cell–cell interactions. Alcohol promotes the release of exosomes from hepatocytes, which are influenced by various cellular processes such as oxidative stress, ER stress, autophagy, and possibly epigenetics. These molecular mechanisms influencing exosome release self-sufficiently cause ALD progression.

## Data Availability

Not applicable.

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
