# Peer review of "Effect of Ethanol on Exosome Biogenesis: Possible Mechanisms and Therapeutic Implications"

_biomolecules, 2023, doi:10.3390/biom13020222_

Round 1

Reviewer 1 Report

The subject is timely and of interest in the field. I have some concerns with regard to the content of the review. Overall the authors do a little overemphasize the role of exosomes in connection with alcohol-associated liver disease. Current knowledge is still limited especially in alcohol associated liver disease field.
- Pathophysiology is complex and they should expand the introduction and at least develop a little more other mechanisms that might be involved (gut barrier dysfunction, microbiota changes, immune mediated mechanisms etc.) including appropriate recent references.

- They should also avoid words like crucial, critical, major etc..

-  Paragraph 5 is a little problematic with a lot of speculation and little hard data especially with regard to liver disease. Maybe point 5.1 and 5.2 are ok but 5.3 and 5.4 need to be revised or even just be summarized in a few words. All the last part of 5.3 dealing with ROS and CYP2E1 would better fit into paragraph 3 (ethanol on exosome release).

- the title is a missleading since the review primarily deals with the impact/link of ethanol on exosomes rather than with hard data showing the connection with alcohol-associated liver disease. Consider revising the title.

Author Response

Reviewer 1:

The subject is timely and of interest in the field. I have some concerns with regard to the content of the review. Overall the authors do a little overemphasize the role of exosomes in connection with alcohol-associated liver disease. Current knowledge is still limited especially in alcohol associated liver disease field.

 Comment. Pathophysiology is complex and they should expand the introduction and at least develop a little more other mechanisms that might be involved (gut barrier dysfunction, microbiota changes, immune mediated mechanisms etc.) including appropriate recent references.

Response. As per the reviewer’s comments, the necessary changes are incorporated into the revised manuscript on page 1, lines 66-79.

Comment. They should also avoid words like crucial, critical, major etc..

Response. The necessary changes are incorporated into the revised manuscript according to the reviewer’s comments.

Comment.  Paragraph 5 is a little problematic with a lot of speculation and little hard data especially with regard to liver disease. Maybe point 5.1 and 5.2 are ok but 5.3 and 5.4 need to be revised or even just be summarized in a few words. All the last part of 5.3 dealing with ROS and CYP2E1 would better fit into paragraph 3 (ethanol on exosome release).

Response.  We appreciate the reviewer for the recommendation and we deleted the first paragraph in Section 5.3 (ethanol, oxidative stress and exosomes), describing basic details about oxidative stress. As we had two articles indicating the link between oxidative stress and exosome release upon ethanol exposure, we retained them in this section. We have also organized this section to better describe the role of oxidative stress on exosome biogenesis upon ethanol exposure. Regarding epigenetics and exosomes, we agree that there is no evidence directly linking ethanol-induced epigenetic changes to altering exosome release. However, the authors believe that compiling evidence available in different disease models/cells would help in addressing the possible role of epigenetics in modulating exosome biogenesis in AALD as ethanol can induce similar epigenetic changes in the liver. We have modified Figure 2 to indicate that this is only a potential mechanism in altering ethanol-induced exosome release.

Comment. The title is missleading since the review primarily deals with the impact/link of ethanol on exosomes rather than with hard data showing the connection with alcohol-associated liver disease. Consider revising the title.

Response. According to the reviewer’s comments, the title of the manuscript is revised as “Effect of ethanol on exosome biogenesis: possible mechanisms and therapeutic implications”

Reviewer 2 Report

The science of extracellular vesicles (EVs) is relatively younger. Currently, a number of laboratories are actively exploring the nature of EVs and it’s connection to different diseases including alcoholic liver diseases (ALD). A comprehensive review on the recent findings concerning the role of EVs in mediating autocrine and paracrine intercellular communication in ALD is in high demand which may help elucidating the potential role of EVs in liver diseases for successful transition from bench to clinic.

The authors, V. Sundar, and V. Saraswathi have summarized the available relevant articles for the role of ethanol induced exosomes in ALD. However, this reviewer believes that addressing following concerns/comments would improve the quality of the article and would be helpful for the readers:

1)      The authors have used abbreviation AALD throughout the manuscript for alcohol-associated liver disease or alcoholic liver disease. ALD is widely used accepted abbreviation for alcohol-associated liver disease/alcoholic liver disease. This reviewer strongly suggests changing AALD to ALD.

2)      The 1st sentence (line 25-26) is not clear and need to be rewritten/rephrased.

3)      The statement in the lines 29-31need to be corrected. Why did the authors avoid cirrhosis in that line?

4)      ‘im-pairs’ should be impairs in line 35

5)      ‘M1 Polarization’ should be ‘M1 polarization

6)      The statement in the line 52 need to be corrected: Exosomes are not isolated entity and are included in EVs. Therefore, correction of the statement is essential to avoid misleading to the readers.

7)      ‘Intraluminal vesicles’ in the line 341 should be ‘intraluminal vesicles’.

8)      Authors summarized some possible mechanisms of alcohol induced exosome release in ALD but did not put in the figures. This reviewer suggests doing so to make readers easy to comprehend the review article.

Author Response

The science of extracellular vesicles (EVs) is relatively younger. Currently, a number of laboratories are actively exploring the nature of EVs and it’s connection to different diseases including alcoholic liver diseases (ALD). A comprehensive review on the recent findings concerning the role of EVs in mediating autocrine and paracrine intercellular communication in ALD is in high demand which may help elucidating the potential role of EVs in liver diseases for successful transition from bench to clinic.

The authors, V. Sundar, and V. Saraswathi have summarized the available relevant articles for the role of ethanol induced exosomes in ALD. However, this reviewer believes that addressing following concerns/comments would improve the quality of the article and would be helpful for the readers:

Comment. The authors have used abbreviation AALD throughout the manuscript for alcohol-associated liver disease or alcoholic liver disease. ALD is widely used accepted abbreviation for alcohol-associated liver disease/alcoholic liver disease. This reviewer strongly suggests changing AALD to ALD.

Response. According to the reviewer’s suggestion, the abbreviation of alcohol-associated liver disease AALD is changed to ALD throughout the revised manuscript.

Comment. The 1st sentence (line 25-26) is not clear and need to be rewritten/rephrased.

Response. According to the reviewer’s suggestion, the necessary changes are made in the revised manuscript.

Comment. The statement in the lines 29-31 need to be corrected. Why did the authors avoid cirrhosis in that line?

Response. We thank the reviewer for pointing it out to include “cirrhosis” and it is included in line 33 on page 1 of the revised manuscript.

Comment. ‘im-pairs’ should be impairs in line 35

Response. In line 38, the necessary corrections are made in the revised manuscript as per the reviewer’s suggestion.

Comment. ‘M1 Polarization’ should be ‘M1 polarization

Response. In line 72, the ‘M1 Polarization’ has been changed to ‘M1 polarization in the revised manuscript as per the reviewer’s suggestion

Comment. The statement in the line 52 need to be corrected: Exosomes are not isolated entity and are included in EVs. Therefore, correction of the statement is essential to avoid misleading to the readers.

Response. The recommended corrections are made on page 2, line 80 of the revised manuscript as per the reviewer’s suggestion.

Comment. ‘Intraluminal vesicles’ in the line 341 should be ‘intraluminal vesicles’.

Response. The recommended changes are made in the figure 2 legend, page 9, line 468 of the revised manuscript as per the reviewer’s suggestion.

Comment. Authors summarized some possible mechanisms of alcohol induced exosome release in ALD but did not put in the figures. This reviewer suggests doing so to make readers easy to comprehend the review article.

Response. The recommended changes including AMPK, sirtuins, and other mechanisms are made in figure 2 of the revised manuscript as per the reviewer’s comment.

Reviewer 3 Report

Studies of exosomes in liver disease, particularly AALD, are rapidly developing. The pathogenesis mechanism of AALD is unclear, but as mechanistic research continues in-depth, experts have realized that exosomes are crucial in regulating intercellular communication during AALD development. In light of the increasing interest in liver pathobiology and exosome biology, the review is timely. The authors have done a wonderful job summarizing the role of ethanol in exosome release and the intracellular processes (autophagy, ER stress, oxidative stress and Epigenetic alternations) involved in AALD-associated exosome biogenesis and release.

Overall, the review is well written, logical and covers the fundamentals and role of exosomes in AALD. As presented, the Figures summarize the current concepts and knowledge available in the literature. However, the manuscript could be strengthened by the following minor revisions.

1) Figure 1: Replace low-quality images with high-resolution images. It is also necessary to modify the figure: Conceptually, exosomes are not only involved in communication between parenchymal cells and NPCs, but also among NPCs, and these events could be crucial in both physiological and pathological processes.

2) Rephrase lines 226-228 to make them more clear

3) Lines 280-282 are confusing. In what way? Please elaborate?

Author Response

Studies of exosomes in liver disease, particularly AALD, are rapidly developing. The pathogenesis mechanism of AALD is unclear, but as mechanistic research continues in-depth, experts have realized that exosomes are crucial in regulating intercellular communication during AALD development. In light of the increasing interest in liver pathobiology and exosome biology, the review is timely. The authors have done a wonderful job summarizing the role of ethanol in exosome release and the intracellular processes (autophagy, ER stress, oxidative stress and Epigenetic alternations) involved in AALD-associated exosome biogenesis and release.

Overall, the review is well written, logical and covers the fundamentals and role of exosomes in AALD. As presented, the Figures summarize the current concepts and knowledge available in the literature. However, the manuscript could be strengthened by the following minor revisions.

Comment. Figure 1: Replace low-quality images with high-resolution images. It is also necessary to modify the figure: Conceptually, exosomes are not only involved in communication between parenchymal cells and NPCs, but also among NPCs, and these events could be crucial in both physiological and pathological processes.

Response. The figures are provided at higher resolution and quality in the revised manuscript as per the reviewer’s comment. Figure 1 is modified to clarify the role of exosomal communication among non-parenchymal cells and parenchymal cells as well in regulating physiological/pathological processes.

Comment. Rephrase lines 226-228 to make them more clear.

Response. Thank you for making this point.  We have revised the whole paragraph to better describe the connection between autophagy and exosome release (Lines 303-314) on page 6 of the revised manuscript.

Comment. Lines 280-282 are confusing. In what way? Please elaborate?

Response. The authors agree with the reviewer’s comment regarding the statement; “Therefore, the cells deploy exosomes as an effective tool to assist with overcoming their antioxidant deficiency and protect themselves from oxidative stress-induced cytotoxicity”. To clarify this, we have included the following sentences with references on page 7, lines 380-382 of the revised manuscript:

“For example, EVs can serve as an alternative mechanism to remove oxidized proteins after oxidative stress to maintain cellular homeostasis [76]. Oxidized lipids are also loaded onto exosomes released from cells undergoing oxidative stress [77]”

Round 2

Reviewer 1 Report

Overall the review is more balanced.

In the new paragraph in the introduction, I am not sure that the cited references are correct. There are many reviews or good papers out in the literature that could be cited with regard to micobiota changes, gut barrier dysfunction etc. This needs to be checked and corrected.

Author Response

Reviewer 1:

Overall the review is more balanced.

Comment.

In the new paragraph in the introduction, I am not sure that the cited references are correct. There are many reviews or good papers out in the literature that could be cited with regard to micobiota changes, gut barrier dysfunction etc. This needs to be checked and corrected.

Response.

The authors thank the reviewer for the comment to improve the manuscript. According to the reviewer’s comment, appropriate references are included in page 2, lines 51-62 of the revised manuscript.
